# *L-carnitine* and *Ginkgo biloba* Supplementation In Vivo Ameliorates HCD-Induced Steatohepatitis and Dyslipidemia by Regulating Hepatic Metabolism

**DOI:** 10.3390/cells13090732

**Published:** 2024-04-23

**Authors:** Amany E. Nofal, Hind S. AboShabaan, Walaa A. Fadda, Rafik E. Ereba, Sherin M. Elsharkawy, Heba M. Hathout

**Affiliations:** 1Zoology Department, Faculty of Science, Menoufia University, Shebin El-Kom 32511, Egypt; nofal83@yahoo.com; 2Clinical Pathology Department, National Liver Institute Hospital, Menoufia University, Shebin El-Kom 32511, Egypt; drhindsaad@liver.menofia.edu.eg; 3Human Anatomy and Embryology Department, Faculty of Medicine, Menoufia University, Shebin El-Kom 32511, Egypt; faddawalaa@gmail.com; 4Department of Pharmacology, Faculty of Medicine, Al-Azhar University, Cario 11511, Egypt; rafikelsayed2015@yahoo.com; 5Physiology Department, Medicine College, Banha University, Banha 13511, Egypt; sherinesharkawy1981@gmail.com; 6Natural Resources Department, Faculty of African Postgraduate Studies, Cairo University, Giza 12613, Egypt

**Keywords:** β-catenin, GLP-1R, hyperlipidemia, insulin resistance, metabolism, natural supplements

## Abstract

Treatment strategies for steatohepatitis are of special interest given the high prevalence of obesity and fatty liver disease worldwide. This study aimed to investigate the potential therapeutic mechanism of L-carnitine (LC) and *Ginkgo biloba* leaf extract (GB) supplementation in ameliorating the adverse effects of hyperlipidemia and hepatosteatosis induced by a high-cholesterol diet (HCD) in an animal model. The study involved 50 rats divided into five groups, including a control group, a group receiving only an HCD, and three groups receiving an HCD along with either LC (300 mg LC/kg bw), GB (100 mg GB/kg bw), or both. After eight weeks, various parameters related to lipid and glucose metabolism, antioxidant capacity, histopathology, immune reactivity, and liver ultrastructure were measured. LC + GB supplementation reduced serum total cholesterol, triglyceride, low-density lipoprotein cholesterol, glucose, insulin, HOMA-IR, alanine transaminase, and aspartate transaminase levels and increased high-density lipoprotein cholesterol levels compared with those in the HCD group. Additionally, treatment with both supplements improved antioxidant ability and reduced lipid peroxidation. The histological examination confirmed that the combination therapy reduced liver steatosis and fibrosis while also improving the appearance of cell organelles in the ultrastructural hepatocytes. Finally, the immunohistochemical analysis indicated that cotreatment with LC + GB upregulated the immune expression of GLP-1 and β-Cat in liver sections that were similar to those of the control animals. Mono-treatment with LC or GB alone substantially but not completely protected the liver tissue, while the combined use of LC and GB may be more effective in treating liver damage caused by high cholesterol than either supplement alone by regulating hepatic oxidative stress and the protein expression of GLP-1 and β-Cat.

## 1. Introduction

Hepatic steatosis, or fatty liver, is a gastrointestinal disorder marked by excessive lipid accumulation in the liver (>5% by weight) induced by alcohol or nonalcoholic reasons [1]. Dyslipidemia is a metabolic disorder characterized by elevated blood levels of low-density lipoprotein cholesterol (LDL-C), total cholesterol (TC), and triglycerides (TGs), accompanied by a reduction in high-density lipoprotein-cholesterol (HDL-C) levels [2]. Dyslipidemia leads to nonalcoholic fatty liver disease (NAFLD), including cirrhosis, mild steatosis, and nonalcoholic steatohepatitis (NASH) [3]. A high-cholesterol diet (HCD) promotes lipid accumulation in the liver, leading to both microvesicular and macrovesicular steatosis and triggering inflammation, which, when combined with obesity, results in a reduction in the body’s ability to defend against excessive reactive oxygen species (ROS) production-induced oxidative stress [3]. The progression from simple steatosis to steatohepatitis is significantly influenced by the accumulation of cholesterol rather than triacylglycerol [4]. Maintaining a diet in which supplements are abundant enhances overall health and guards against a range of disorders [5].

L-carnitine (LC), a quaternary ammonium molecule (β-hydroxy-γ-N-trimethylaminobutyric acid), is synthesized from L-lysine and L-methionine in the kidneys and liver. It exerts various physiological effects by participating in energy metabolism [6,7,8]. Its primary role involves serving as an essential mitochondrial respiratory cofactor, facilitating long-chain fatty acid transport into the mitochondrial matrix for β-oxidation, thereby increasing energy production [9]. LC not only functions as a transporter in fatty acid β-oxidation but also exhibits numerous beneficial functions, including increasing glutathione levels, reducing liver lipid peroxidation [10], decreasing hepatic inflammatory cytokines [11], ameliorating hepatic dysfunction induced by a high-fat diet [8], and suppressing heart fibrosis [12]. Additionally, LC effectively normalized insulin sensitivity in type 2 diabetic patients by regulating the synthesis of key glycolytic and gluconeogenic enzymes [13]. Scientific interest persists in exploring LC as a potential therapy for kidney disease, cardiovascular issues, diabetes, and symptoms related to carnitine deficiency and mitochondrial disorders [14,15,16].

There is a pressing necessity to explore alternative natural compounds derived from medicinal plants, herbs, and spices. This is due to the numerous unwanted side effects associated with synthetic chemical drugs used to treat metabolic disorders and obesity. *Ginkgo biloba* (GB, G. biloba: Syn.: *Salisburia adiantifolia*) has been used by humans for more than 2000 years as a valuable herb and is regarded as a living fossil [17]. It possesses a rich history in traditional medicine and has been demonstrated to offer therapeutic advantages, encompassing anti-inflammatory, photoprotective, hepatoprotective, cardioprotective, and antioxidant properties [17]. The ability of *Ginkgo biloba* leaf extract to protect organs from damage was attributed to its active constituents; terpenoids, such as *Sesquiterpene bilobalide*; diterpenoids A, B, C, M, and J ginkgolides; flavonoid glycosides, such as quercetin, kaempferol, and isorhamnetin; proanthocyanidins; biflavones; alkylphenols; 4-O-methylpyridoxine; simple phenolic acids; 6-hydroxykynurenic acid; and polyprenols [18]. Once GB is administered orally or parenterally, ginkgolide A and B, flavonoids, and biolalide become available [19]. Flavonoid glycoside extracts enhance hepatic steatosis by inhibiting lipid absorption and hepatic lipogenesis and improving hepatic fatty acid oxidation, which leads to an improvement in dyslipidemia [20]. The antioxidant defense-enhancing and anti-inflammatory properties of flavonoids provide protection against HDL dysfunction and cardiovascular disorders in the context of inflammatory disease conditions, such as atherosclerosis or obesity [21]. Additionally, Sikder et al. [22] reported the protective effect of common flavonoids against hepatotoxicity and inflammation induced by a high-cholesterol diet (2%), as increases in body weight, liver function enzymes, lipid levels, lipid peroxidation levels, and the expression of markers of inflammation were significantly prevented when quercetin or rutin was taken along with an HCD. When GB was used to treat hyperlipidemia, it showed adaptive and regulatory effects [23]. Because GB can neutralize ferryl ion-induced peroxidation and scavenge free radicals, its activity has been linked to the treatment and prevention of diseases related to oxidative stress [24].

L-carnitine and *Ginkgo biloba* are commercial products widely used as nutraceutical herbs. According to many previous studies, LC and GB have antioxidant, anti-inflammatory, hypolipidemic, and anti-obesity effects that have beneficial effects on insulin sensitivity, protein nutrition, and dyslipidemia [8,22,24,25]. In various animal models, LC and GB decreased body weight and visceral fat accumulation and accelerated food intake normalization [8,25,26,27]. As a result, they cause weight loss and hypoglycemia by increasing insulin sensitivity and decreasing insulin resistance by lowering free fatty acid levels or regulating cellular energy metabolism. GB and LC are healthy combinations that regulate the brain health status of kindled rats, and they significantly relieve alterations in the levels of monoamines in the hypothalamus and hippocampus of rats, reflecting the powerful anticonvulsant effects of LC and GB [26,27,28]. To the best of the authors’ knowledge, there are no reports on the role of the combination of LC and GB in modulating HCD-induced hyperlipidemia. Therefore, this study sought to assess the underlying mechanisms through which supplementation with L-carnitine, *Ginkgo biloba*, or their combination mitigates HCD-induced hyperlipidemia and liver steatosis.

## 2. Materials and Methods

### 2.1. Materials

The standard rodent diet and high-cholesterol diet were procured from the Egyptian Company of Oils and Soap in Kafr-Elzayat, Egypt. L-carnitine (1 g in each tablet) and *Ginkgo biloba* capsules were obtained from Health Shop (an online Puritan’s Pride Products Store in Egypt, support@ths-egypt.com), and each GB hard gelatin capsule contained 260 mg of GB leaf extract powder, which was standardized to 24% ginkgo flavone glycoside and 6% total ginkgolide (terpene lactones).

### 2.2. Animals and Treatment

Adult male Wistar albino rats weighing between 130–145 g (9–10 weeks old) were purchased from the experimental animal facility in Helwan, Egypt. The care of the rats and all experimental procedures were conducted in compliance with institutional animal ethics guidelines and sanctioned by the Institutional Animal Care and Use Committee of Menoufia University under approval No. MUFS/F/HI/1/22. Animal welfare was ensured by maintaining the animals in a controlled environment with a temperature ranging from 25 ± 5 °C and humidity between 50–70% under a 12 h light/day cycle. Five groups of fifty rats were randomly allocated, each consisting of 10 rats, and were observed for nine weeks (one week for adaptation and 8 weeks for trial duration) [29]. The rats in each group were assigned to two cages (*n* = 5 for each cage). The control group (G1) was fed standard commercial chow (100 g) containing 76% wheat, 5% corn starch, 10% casein, 4.7% cellulose, 1% vitamin mixtures, and 3.3% mineral mixtures. The remaining four groups were fed a high-cholesterol diet (HCD) to induce fatty liver. The prepared HCD (100 g) contained 17.48% protein, 52.99% carbohydrates, 10% cholesterol, 6.85% fat, 4.08% ash, and 2.16% vitamins and minerals [30]. The second group (G2) received only an HCD without any further treatment. The remaining three groups received an HCD along with daily oral administration of either LC (G3; 300 mg LC/kg bw), GB (G4; 100 mg GB/kg bw) [27,28], or both LC + GB (G5) at the same previous doses.

### 2.3. Serum Biochemical Analysis

After the 8-week experiment concluded, animals that had fasted overnight were sedated with 100% isoflurane (2 mg/kg body weight or equivalent to 2% inhaled concentration) and dissected. Using a dry Eppendorf tube, blood was collected via cardiac puncture, allowed to coagulate for 30 min at room temperature, and then centrifuged for 10 to 15 min at 3000 rpm.

Nonhemolyzed serum samples were promptly collected and stored at −20 °C for the evaluation of various parameters, including total cholesterol (TC), triglyceride (TG), high-density lipoprotein (HDL-C), low-density lipoprotein (LDL-C), fasting blood glucose (FBG), insulin, homeostatic model assessment for insulin resistance (HOMA-IR), aspartate aminotransferase (AST), and alanine aminotransferase (ALT) levels. The levels of TC, TG, HDL-C, LDL-C, AST, and ALT were determined using biochemical assays following the instructions provided with the corresponding kits (Cat. No. A110-1-1, A111-1-1, A112-1-1, A113-1-1, C009-3-1, and C010-3-1, respectively, Nanjing Jiancheng Bioengineering Institute, Nanjing, China). Additionally, FBG levels were measured using a kit according to the manufacturer’s instructions (Cat. No. YC0713; Shanghai YaJi Biotechnology Co., Ltd., Shanghai, China). These assays were conducted by using an automatic biochemical analyzer (AU480 from Beckman Coulter, Miami, FL, USA).

### 2.4. Antioxidant Markers

The analysis of malondialdehyde (MDA), superoxide dismutase (SOD), and catalase (CAT) activity was conducted spectrophotometrically utilizing a UNICO UV-2100 spectrophotometer according to Ohkawa et al. [31], Nishikimi et al. [32], and Aebi [33], respectively. Commercial kits provided by Bio-Diagnostics Co., Giza, Egypt, were used for these analyses following the manufacturer’s instructions (Cat. No. MDA: MD 2529, SOD: SD 2521, CAT: CA 2517).

### 2.5. Histological and Immunohistochemical Techniques

Following animal dissection, liver samples were promptly fixed in 10% neutral formalin. Sections were prepared using the conventional paraffin embedding method, and histological investigation was performed under a light microscope using 4–5 μm thick sections stained with hematoxylin and eosin (H. and E.) and Masson’s trichrome [34,35]. Immunohistochemical staining was performed on paraffin-embedded liver tissue sections. Primary antibodies against the glucagon-like peptide-1 receptor (GLP-1R, 1:200, NBP1-97308, Novus Biologicals, USA) and beta-catenin (β-Cat, BD Biosciences, San Jose, CA) were used as described by Jensen et al. [36] and Zou et al. [37], respectively. Subsequently, the slides were exposed to 3,3’-diaminobenzidine tetrahydrochloride (DAB) substrate from Santa Cruz, USA, followed by washing in Tris-buffered saline (TBS) and counterstaining with hematoxylin.

### 2.6. Histological Lesion Scoring and Image Analysis

The quantitative grading and scoring of histological lesions were conducted through microscopic examination of H. and E.-stained liver sections, assessing 10 random fields within each microscopic power field (MPF) at 20× magnification. A quantitative scoring system for evaluating nonalcoholic steatohepatitis was used with slight modifications [38]. The intensity of liver fibrosis was evaluated on Masson trichrome-stained sections by the modified Knodell scoring system [39]. Semiquantitative scoring of the percentage of blue-stained collagen deposition areas with Masson’s trichrome and the brown-stained immunohistochemical expression of GLP-1R and β-Cat was performed on 10 random digital images from high microscopic power fields (HPFs; 40×) and then analyzed via ImageJ software system (Java-based application for analyzing images, 1.51j8, USA) [40].

### 2.7. Ultrastructure Study

Tiny liver tissue samples measuring 1 mm^3^ were first preserved in a fixative solution containing formalin and glutaraldehyde (4F1G) in phosphate-buffered saline. Subsequently, they were postfixed with 2.0% osmium tetroxide and embedded in resin, as described by Hayat [41]. Semithin sections were produced and dyed with toluidine blue for observation using a light microscope. For ultrathin sections, specific areas from the semithin sections were chosen, placed on copper grids, and treated with uranyl acetate and lead citrate for staining. These ultrathin sections were analyzed using a Jeol Japan transmission electron microscope (Jem-1400) at the Electron Microscope Unit, Alexandria University, Egypt.

### 2.8. Statistical Analysis

The data are presented as the mean ± standard deviation (SD). Statistical significance was determined through one-way ANOVA, and subsequently, the Tukey post hoc test was employed for multiple comparisons. The statistical analysis was carried out using SPSS software (version 25.0). *p*-values less than 0.05 were considered to indicate statistical significance.

## 3. Results

### 3.1. Body Weight

The current data in Table 1 show that body weight is significantly greater in the HCD group (G2), by approximately 64.79%, than in the control group (G1). Compared with those in the control group (G1), the percentages in the HCD + LC (G3), HCD + GB (G4), and HCD + LC + GB (G5) treatment groups are significantly greater; the percentages are ~38.8% for G3, 44.6% for G4, and 21.67% for G5. Compared with those in the HCD group (G2), the percentages in all the treated groups are notably lower; compared with those in the HCD group, the percentages are 15.78% lower for G3, 12.25% lower for G4, and 26.17% lower for G5.

### 3.2. Biochemical Parameters

#### 3.2.1. Glucose Homeostasis

Investigations were conducted across all groups to evaluate fasting blood glucose (FBG), insulin, and HOMA-IR (Table 1). The HCD-fed rats have approximately 60.4%, 29.6%, and 36.7% greater levels of FBG, insulin, and HOMA-IR, respectively, than the control rats. Furthermore, treatment with only LC or GB has a significant effect on FBG, which changes by approximately 13.91% and 13.71%, respectively, and on HOMA-IR by 4.31% and 6.27%, respectively, compared with those in HCD-fed rats. However, insulin does not significantly change in these treated groups by approximately 12.82% and 23% in the HCD group. Moreover, in the groups treated with both LC and GB, the FBG, insulin, and insulin resistance indices are significantly lower (*p* < 0.05) than those in the HCD group by approximately 31.44%, 29.5%, and 19.86%, respectively.

#### 3.2.2. Lipid Profile

The findings from the serum lipid profile analysis revealed that treatment with LC or GB nearly normalizes the lipid profile (TC, TG, HDL-C, and LDL-C) of HCD-fed rats (Figure 1). Compared with those in the control group, the HCD-fed group exhibited notable increases in the serum levels of TC, TG, and LDL, with increases of 26.6%, 42.8%, and 62.69%, respectively. However, the serum level of HDL-C is 15.3% lower in HCD-fed rats than in control rats. Treatment of HCD-fed rats with both LC and GB decreases the serum levels of TC, TG, and LDL-C and increases the serum HDL-C. The cholesterol, triglyceride, and LDL-C levels are reduced by approximately 2.58%, 27.44%, and 7.15%, respectively, in the HCD + LC group and by 22.03%, 37.73%, and 16.45%, respectively, in the HCD + GB group compared with those in the HCD group. Additionally, the level of HDL-C increased by 36.82% in the HCD + LC group and by 65.55% in the HCD + GB group compared with that in the HCD group. The results of the HCD + LC + GB treatment group indicate that this treatment reduces the serum levels of TC, TG, and LDL-C to levels close to those of the control group, while the level of HDL-C significantly increases by 52.3% compared with that of the control group.

#### 3.2.3. Liver Enzymes

Compared with those in the control group, the ALT and AST levels in the serum of HCD-fed rats without any therapy are significantly (*p* < 0.05) elevated by approximately 90.31% and 75.66%, respectively (Table 1). On the other hand, ALT and AST are significantly lower in HCD-fed and HCD-treated animals than in HCD-fed animals (31.82% and 28.92% for HCD + LC, 32.37% and 28.72% for HD + GB, and 40% and 31.93% for HCD + LC + GB, respectively).

### 3.3. Antioxidant Markers

In HCD-fed rats, treatment with LC and GB resulted in a significant increase in the levels of the enzymatic antioxidants SOD and CAT and a significant decrease in the levels of the antioxidant stress marker malondialdehyde (MDA). The SOD, CAT, and MDA data are summarized in Figure 2. The activities of antioxidant enzymes (SOD and CAT) and MDA were measured in the control and exposed groups. The activities of antioxidant enzymes (SOD and CAT) are significantly lower in the HCD group (~55.46% and ~94.57%, respectively) than in the control group. However, CAT and SOD activities are not significantly elevated in the HCD + LC, HCD + GB, and HCD + LC + GB groups (2.17%, 7.75%, 3.27%, and 2.96%, 2.91%, 1.17%, respectively). In HCD-fed animals, the MDA concentration in the serum significantly (*p* < 0.05) increased by approximately 494.2% compared with that in the control group. Compared with those in the HCD group, the percentages in the HCD + LC group, HCD + GB group, and HCD + LC + GB group are ~24.26%, 40.67%, and 46%, respectively.

### 3.4. The Histological Observations

Liver H. and E. sections of the control rats exhibited typical hepatic parenchyma of classic lobules with radially arranged polyhedral hepatocytes and blood sinusoids lined by endothelium and some von Kupffer cells (Figure 3a). After 8 weeks of HCD feeding, liver sections revealed multiple histopathological lesions throughout the hepatic parenchyma, including microvesicular and macrovesicular hepatic steatosis, balloon degeneration, inflammation, focal necrotic areas with pyknotic nuclei, and abundant apoptosis. Hepatocellular degeneration occurred with enlarged vacuolated hepatocytes, and the nuclei were displaced toward the cell periphery by diffuse intracytoplasmic fat globules. In addition, portal hepatitis and mononuclear inflammatory cell infiltration around dilated congested blood vessels and proliferated bile ducts of the portal areas were frequently observed (Figure 3b,c). However, liver sections from animals fed HCD supplemented with LC or/and GB for 8 weeks exhibited noticeable alleviation of these histopathological features (Figure 3d–f). LC or GB mono-treatment resulted in mild hepatocellular vacuolation, dilated sinusoids, increased Kupffer cells, and few pyknotic nuclei scattered throughout the tissue (Figure 3d,e). All histopathological lesions became less prominent and more pronounced in livers obtained from rats cotreated with both LC and GB, where the hepatocytes, blood sinusoids, and central veins retained a normal appearance similar to that of the control animals (Figure 3f).

In this study, another noteworthy histological finding was the presence of portal fibroplasia in Masson’s trichrome sections, characterized by a notable increase in the percentage of blue collagen fibers extending from the portal tracts into the liver parenchyma, indicating the development of hepatic fibrosis in rats fed an HCD (G2) in comparison with those in the control group (G1), which were fed a standard diet (Figure 4a–c). Mild fibroblastic proliferation and collagen fiber deposition were restricted to the portal area in liver sections from HCD-fed rats and HCD-fed rats monocreated with LC (G3) (Figure 4d). Marker improvement with less evidence of fibroblastic proliferation and collagen fiber deposition was recorded in the rats fed HCD and treated with GB alone or with both LC and GB (G4 and G5) (Figure 4e,f). Compared with that in the control group, the percentage of collagen fibers in the HCD-treated rats is markedly greater (Figure 4). Conversely, HCD-treated rats treated with either GB alone, LC alone, or a combination of both LC and GB demonstrate a substantial decrease in collagen fiber content, with reductions of approximately 66.28%, 74.73%, and 89.25%, respectively, in comparison with those in the HCD group. Table 2 shows the results from the histological quantitative grading and scoring system of the previous histological lesions from rat livers of all the current groups.

### 3.5. Immunohistochemical Observations

Figure 5 and Figure 6 show the protein expression of membranous β-Cat and cytoplasmic GLP-1 in immunohistochemical liver sections from the study groups. Livers from the control group displayed strong membranous immune expression of β-Cat, including cell borders, and cytoplasmic expression of GLP-1 in parenchymal hepatocytes (Figure 5a and Figure 6a, respectively). Compared with those in the control group, immune reactivity to β-Cat and GLP-1 is significantly decreased in the HCD-fed rats by approximately 59.84% and 47.57%, respectively (Figure 5b and Figure 6b, respectively). Moreover, immune reactivity to β-Cat and GLP-1 is restored in HCD-fed rats and those monocreated with LC or GB (Figure 5c,d and Figure 6c,d, respectively). Compared with those in HCD-fed rats, β-Cat and GLP-1 in HCD + LC, HCD + GB, and HCD + LC + GB-fed rats are higher (27.24%, 88.22%, 104.63% for β-Cat and 49.65%, 63.21%, 68.04% for GLP-1). In comparison, livers from HCD-fed rats that received both LC and GB displayed significantly increased immunoreactivity to β-Cat and GLP-1, similar to that of the control rats (Figure 5e and Figure 6e, respectively).

### 3.6. Ultrastructure Alterations

Transmission electron microscopic examination of liver sections from control rats revealed a typical hepatic ultrastructure, where hepatocytes exhibited normal nuclei, rough endoplasmic reticulum, a Golgi apparatus, and mitochondria with well-developed cristae scattered within the cytoplasm. In addition, bile canaliculi with short microvilli were present between hepatocytes (Figure 7a,b). Rats fed an HCD for 8 weeks exhibit several dramatic ultrastructural alterations, with degenerated cytoplasmic organelles, including irregular cell membranes, abnormal small marginal nuclei with dispersed chromatin and prominent nucleoli, a rarified cytoplasm with reduced glycogen granules, a dilated rough endoplasmic reticulum, multiple lipid vacuoles of variable size, and degenerated, dense, fused, and pleomorphic mitochondria with partial or complete loss of cristae (Figure 7c,d). Ultrathin sections of livers obtained from rats fed an HCD supplemented with LC or GB for 8 weeks exhibited noticeable alleviation of these ultrastructural alterations (Figure 7e–h), and some abnormalities, such as dilated rough endoplasmic reticulum, small vacuoles, and few lysosomes, were still observed after LC or GB mono-treatment (Figure 7e–f). The LC and GB dual-treatment group displayed an advanced degree of improvement, and the restored glycogen granules and cell organelles seemed to be seminormal (Figure 7g,h).

## 4. Discussion

A high-cholesterol diet disturbs the balance between lipid processing and absorption, leading to disturbances in lipid metabolism. These disruptions are particularly prevalent in obese people and can lead to various metabolic disorders, such as diabetes, fatty liver disease, high blood pressure, and atherosclerosis [42]. The current study demonstrated that rats fed an HCD for 8 weeks exhibited increased weight gain and fatty liver compared with those fed a standard diet due to increased caloric intake and increased energy expenditure, which subsequently led to adipose tissue accumulation. However, the body weights of rats in the different treatment groups (HCD + LC, HCD + GB, and HCD + LC + GB) were significantly lower than those of rats in the HCD group without any treatment. These findings agree with previous studies demonstrating that rats fed an HCD experience significant weight gain, which is indicative of obesity, within a period ranging from four to eight weeks [8,43]. LC supplementation in mice fed an HFD without exercise resulted in a significant reduction in weight gain [44]. LC and GB treatments balanced the increase in body weight in different animal models. Notably, in this study, TG, TC, and LDL-C levels were significantly elevated in the HCD group compared with those in the control group. Conversely, the concentration of HDL-C decreased significantly, indicating potential lipid-related metabolic issues. However, treatment with LC and/or GB effectively inhibited these increases (Figure 1) and significantly reduced the serum TG, TC, and LDL-C levels, which were close to their levels in the control group. Additionally, the levels of LDL-C and HDL-C, which are important indicators of nonalcoholic fatty liver disease (NAFLD) progression, were altered by an HCD but normalized with LC and/or GB treatment. Obese female rats exhibited a significant increase in the serum levels of TC, TG, VLDL, and LDL-C, along with a notable reduction in HDL-C levels [45,46]. Abnormal elevations in TC and TG have been linked to NAFLD development [25], and hypercholesterolemia is considered a risk factor for liver injury [47]. Excessive triglyceride deposition within hepatocytes leads to liver lipotoxicity and impaired fatty acid oxidation, contributing to NAFLD progression [48]. Cholesterol and triglycerides play crucial roles in metabolism, acting as energy sources, membrane components, and precursors for various biological molecules. Monitoring their levels in blood samples is clinically significant due to their implications for liver health and overall metabolic function [49]. Moreover, Fidèle et al. [50] reported that the accumulation of cholesterol resulted in an increase in the production of steroid hormones, including cortisol, estrogens, and testosterone. This hormonal increase in rats that were fed a high-fat diet was associated with weight gain. LDL-C levels increase in HCD-fed rats. LDL-C is a poor cholesterol, where the level of fat is greater than the level of protein, and transports cholesterol from the liver to all parts of the body. LDL-C is a significant component of cholesterol related to a greater risk of atherosclerosis, as it serves as a physiological carrier for cholesterol distribution to peripheral tissues and especially deposits it on the walls of blood vessels, causing an increase in blood clots; in contrast, HDL-C is known as good cholesterol, where the level of protein is greater than the level of the fat itself and transports cholesterol from different parts of the body to the liver for recycling and excretion in bile [49]. Treatment with LC altered hepatic lipid metabolism potentially through carnitine-mediated lipid metabolism, modulation of hyperglycemia, and enhancement of self-antioxidant capacity. This intervention resulted in reduced serum lipid levels, triglycerides, glucose, and liver enzymes but also improved the cholesterol profile compared with that of rats fed a high-cholesterol diet in the present study. Similarly, rats fed an HFD had a high level of LDL (153.33 ± 1.38 mg/dl) but a low serum level of HDL (40.60 ± 1.16 mg/dL) [51]. Several studies support these observations. Kim et al. [52] reported the beneficial effects of LC in lowering lipid levels in both the blood and liver, as they highlighted its role in promoting fat oxidation, thus reducing its accumulation. Similarly, Mayes et al. [53] suggested that LC functions as a fat burner by optimizing fat oxidation, thereby limiting its storage. Research by González-Ortiz et al. [54] demonstrated that supplementing obese rats with LC led to a reduction in the serum levels of TC, TG, LDL, free fatty acids, and very low-density lipoprotein (VLDL), while it particularly increased the level of HDL. Furthermore, LC was found to ameliorate fatty liver, dyslipidemia, and hepatitis by modulating lipid metabolism, antioxidant capacity, and inflammatory responses, as noted by Su Chang Chao and colleagues [55]. Additionally, Zhang et al. [23] revealed that GB has a multifaceted impact on the lipid profile of the rat metabolome. It regulates polyunsaturated fatty acids, limits cholesterol absorption, and deactivates 3-hydroxy-3-methylglutaryl–coenzyme A (HMG-CoA). A meta-analysis by Fan et al. [56] suggested that combining GB with statin therapy improved TC and TG and enhanced HDL-C compared with statin therapy alone. Moreover, in a study involving male rabbits, Hussein et al. [45] reported that, compared with no treatment, GB treatment significantly reduced plasma TG and cholesterol levels while increasing HDL-C levels. These findings align with the results presented in Figure 1 of the current study.

*Ginkgo biloba* and L-carnitine have also provided substantial advantages in conventional medicine, such as promoting weight loss and exhibiting antidiabetic, antihypertensive, and antilipidaemic effects [13,43,57,58]. These attributes hold promise for treating metabolic syndrome and reducing the heightened risk of cardiovascular events [59]. In this HCD-fed rat experiment, metabolic dysregulation characterized by hyperglycemia, hyperinsulinemia, and elevated insulin resistance (HOMA-IR) was observed, potentially stemming from increased energy storage in triglycerides and elevated circulating free fatty acid levels due to high cholesterol intake, leading to insulin resistance [8]. The heightened blood glucose levels may result from reduced insulin efficacy in controlling hepatic glucose production [57]. Treatment with either LC or/or GB significantly (*p* < 0.05) reduced the fasting blood glucose, insulin, and insulin resistance indices compared with those in the HCD group, consistent with the findings of Jing et al. [60] for GB. Disruptions in glucose regulation were associated with significant alterations in the serum lipid profile. Similarly, rats fed an HFD had significantly greater blood glucose levels and lipid profiles [51]. In a study by Li et al. [25], NAFLD induced by a high-fat diet resulted in hepatic steatosis, lipid accumulation, inflammation, liver injury, glucose intolerance, and insulin resistance in mice. However, treatment with GB significantly improved these conditions, potentially through enhancing insulin receptor substrate 1 (IRS-1) signal activation and reducing nuclear factor kappa B (NF-κB) and endoplasmic reticulum stress (ERS) signal activation. The potential mechanism underlying the antihyperlipidemic effect of GB may involve alterations in the activity of cholesterol biosynthesis enzymes and/or modifications in lipolysis levels, both of which are regulated by insulin [61]. Treatment of HCD-fed mice with LC or GB improved glucose intolerance, insulin resistance, lipid accumulation, hepatic steatosis, and liver injury [62,63].

Lipid peroxidation (MDA) and substantial antioxidant (CAT and SOD) levels were significantly greater in HCD-fed rats than in control rats in this study, indicating elevated free radical levels and increased oxidative stress. Conversely, the HCD + LC, HCD + GB, and HCD + LC + GB groups exhibited a marked reduction in MDA levels and a significant increase in CAT and SOD compared with those in the HCD group. These results suggest that an HCD induces hepatic oxidative stress, potentially due to hyperglycemia in an HCD, which enhances polyol pathway activity and inhibits the pentose phosphate pathway, thereby reducing intracellular NADPH levels. NADPH is essential for regenerating GSH from its oxidized form [64]. The results indicated that LC and GB have protective effects against oxidative damage, likely due to their antioxidant properties and ability to scavenge free radicals. These results align with a study by Tousson et al. [26], which demonstrated a significant decrease in MDA levels and an increase in SOD and CAT activities in the kidneys of rats treated with GB and LC compared with those of control rats. Sharma and Yadav [65] suggested that LC is a successful dietary treatment for improving weakened biochemicals and renal function in chronic kidney disorders. LC and GB have potent protective and therapeutic effects against pentylenetetrazol-induced epilepsy by addressing antiepileptic actions and oxidative imbalances [27,28]. This finding aligns with that of Zhang et al. [66], who demonstrated that GB improved the antioxidant status and reduced free radical-induced lipid peroxidation in the central nervous system of rats fed an HFD. Furthermore, *Ginkgo biloba* was shown to reduce MDA levels and increase GSH levels in aortic tissue [45]. In obese individuals, the presence of ROS leads to oxidative stress [67,68], which, in turn, triggers inflammation, leading to the release of inflammatory factors [69]. The redox and oxidative defense system consists primarily of enzymatic and nonenzymatic antioxidants, which efficiently eliminate ROS and breakdown peroxides [70,71]. The antioxidant system, which includes CAT and SOD, plays a crucial role in scavenging free radicals and safeguarding cells from oxidative stress [72]. The end product of lipid peroxidation (MDA) is very toxic to cells and their membranes [73,74]. Elevated MDA levels serve as an indicator of oxidative state imbalance in newly diagnosed patients [75].

In this study, HCD model rats exhibited increased not only body weight, glucose, insulin resistance, lipid profile, and lipid peroxidation but also elevated liver enzymes (Table 1) and exacerbated deposition of lipid droplets and collagen fibers in their liver sections (Figure 3b,c and Figure 4b,c). However, treatment with LC and/or GB significantly mitigated these effects (*p* < 0.05, Table 2). The lesions observed in the liver tissues of HCD-fed animals in this study included several hepatic histological and ultrastructural disorders, including both portal areas, hepatocytes, progressive enlargement of sinusoids, micro- and macrovesicular fatty degeneration, intracytoplasmic organelles, steatohepatitis and periportal fibrosis. Previous research has demonstrated that feeding rats and rabbits an HCD results in significant lipid accumulation in the liver [58]. An HFD induced several hepatic histopathological and ultrastructural changes in prediabetic rats, as demonstrated by the presence of degenerated cytoplasmic hepatocytes with mitochondrial swelling, endoplasmic reticulum dilation, plasma membrane blebbing, cytoplasmic vacuoles and lipid droplets [8,76]. Two primary histological patterns of hepatic steatosis define fatty disorders. Microvesicular steatosis involves the substitution of cytoplasm with small fat vacuoles while the nucleus remains centrally positioned. On the other hand, macrovesicular steatosis is characterized by the presence of large fat droplets, where the cytoplasm is largely replaced by a sizable fat vacuole that shifts the nuclei towards the cell’s outer edge [77,78]. Similarly, hepatocellular ballooning is a crucial histological indicator for diagnosing NASH. Several studies have verified that the release of liver enzymes into the bloodstream is a significant distinguishing characteristic, indicating an elevated risk of disease progression [79,80]. In this study, LC + GB triggered severe microvesicular steatosis. Moreover, ultrastructural alterations in hepatocytes, including mitochondria, rough endoplasmic reticulum, and cell nuclei, were noted. These results are in accordance with those of Abdel-Emam et al. [81], who concluded that the coadministration of LC with chlorpheniramine maleate or cetirizine hydrochloride induced less severe histopathological alterations in the liver. Additionally, LC alleviated hepatic cord distortion, blood vessel congestion and dilatation, and hepatocyte degeneration induced by letrozole in female rats [82]. Rashad et al. [83] reported the mitigating impact of LC on atrazine-induced hepatotoxicity in rats. These findings are in line with those of Li et al. [25], who reported that *Ginkgo biloba* extract has a potential therapeutic effect on liver injury in HFD-fed rats by attenuating liver inflammation and fibrosis. Consideration should also be given to circumstances in which GB is used as a required therapy because it causes hypothyroidism in rats [84]. These findings confirmed previous data indicating that HCD-induced liver oxidative stress triggers inflammatory and fibrogenic signaling pathways that promote liver steatosis and fibrosis progression. These alterations may be due to increased ROS production associated with an HCD, resulting in chronic inflammatory response stimulation, which is commonly related to NAFLD and may also increase liver damage [85]. Thus, the present study evaluated histological inflammatory markers in the liver tissues of untreated and treated HCD-fed rats. LC and/or GB alleviated lipid accumulation and hepatic steatosis in HCD-fed rats (Figure 3 and Table 2). The elevated levels of the liver enzymes ALT and AST in the serum of rats fed HCD provided further evidence of the impact of LC and/or GB and were considerably reduced (*p* <0.05) following LC and/or GB treatment (Table 1). Elevated AST and ALT serum levels may be caused by inflammation inside liver cells, which releases more proinflammatory cytokines and more severely damages liver cells by injuring hepatocytes [8]. Serum ALT and AST are useful biomarkers of liver injury [86]. Elevated serum levels of ALT and AST were recorded in a modified dyslipidemia model [87]. Other studies revealed that treatment with a high dose of GB or LC significantly reduced the serum levels of ALT and AST markers [88,89]. However, Nobili et al. [90] showed no effect on reducing ALT, inflammation, or steatosis in 53 patients with NAFLD in a two-year clinical trial of patients treated with vitamin C and vitamin E (600 IU/day), which is consistent with current evidence on inflammation but not ALT or steatosis. On the other hand, ursodeoxycholic acid, a drug for primary biliary cirrhosis, along with vitamin E (400 IU twice daily) for the same period (2 years) had no effect on inflammation in 48 patients with advanced nonalcoholic fatty liver disease (NAFLD) but decreased alanine transaminase (ALT) and steatosis [91]. Overall, this study showed that giving rats an HCD for eight weeks caused hepatic steatosis, which harmed hepatocyte ultrastructure and produced liver injury enzymes in the bloodstream, possibly through an increase in lipids and oxidative stress indicators. LC and/or GB were effective at inhibiting liver dysfunction. Furthermore, the current biochemical, histological and ultrastructural data confirmed that combination therapy (HCD + LC + GB) restored hepatic structure, function, and metabolic damage and increased the number of organelles in hepatocytes as a result of an HCD. Based on these outcomes and to correlate hepatic alterations and the related underlying mechanisms, it was necessary to study markers involved in hepatic metabolism. This metabolic dysfunction was immunohistochemically determined in the present study, which indicated that an HCD decreased the protein expression of hepatic GLP-1R and β-catenin, while HCD rats that received LC and/or GB exhibited significantly restored protein expression levels that were similar to those of the control rats. Tveden-Nyborg et al. [92] reported similar hepatic histological and ultrastructural features in dyslipidemic guinea pigs, which were attributed to the breakdown of mitochondrial polyunsaturated lipids, causing cell death and heightened oxidative stress. Similarly, in steatotic rats, Ellatif et al. [76] observed abnormal hepatocyte ultrastructures associated with oxidative stress, manifested as an enlarged rough endoplasmic reticulum, degenerated mitochondria, increased lysosomes, damaged nuclei, and misshapen nuclear membranes. Natural products have demonstrated efficacy in mitigating hepatocellular damage by exerting antioxidative effects [93].

Dysregulated β-catenin is pivotal for abnormal hepatic growth and the development of health issues such as obesity, diabetes, NAFLD, and metabolic syndrome [94]. Altered β-catenin expression is associated with NAFLD and other liver diseases by influencing genes controlling glucose and nutrient metabolism through interactions with transcription factors such as forkhead box protein O, T-cell factor, and hypoxia-inducible factor 1α [95,96]. β-catenin is a critical component of the Wnt signaling pathway and affects cell differentiation and proliferation, thus impacting liver physiology and development. Changes in β-catenin activity are implicated in NAFLD pathogenesis [96,97,98]. The current study utilized immunohistochemistry to identify β-catenin, which is a multifunctional boundary cadherin protein that serves as an intracellular signaling transducer in the WNT signaling pathway) in liver tissue and acts as a hepatic metabolism indicator involved in liver homeostasis and the regulation of cell adhesion, regeneration, proliferation, hypoxia resistance, apoptosis, steatosis, cholesterol metabolism and other biological processes [99], as it regulates the expression of genes that control glucose, xenobiotic, and nutrient metabolism. Changes in β-catenin signaling trigger the stimulation of hepatic stellate cells, causing fibrosis and contributing to nonalcoholic steatohepatitis pathogenesis [96]. Lehwald et al. [100] reported that reagents that affect Wnt-β-catenin signaling might be potential therapeutic options for treating liver disease, as nutrient oxidative stress compromises the function of β-catenin, which plays a crucial role in maintaining homeostasis in mitochondria and controlling ATP synthesis via fatty acid oxidation, oxidative phosphorylation (OXPHOS), and the tricarboxylic acid cycle. LC promoted neurogenesis in mesenchymal stem cells via the Wnt-β-catenin signaling pathway [101]. Furthermore, immunohistochemical analysis of GLP-1R (glucagon-like peptide-1 receptor) in liver sections in the present study played a critical role in clarifying the beneficial effect of the combined treatment of LC and GB on HCD-fed rats. GLP-1R is secreted by intestinal L cells and acts as a metabolic regulator, influencing processes in the liver, intestine, and brain, and is associated with weight loss promotion and improved insulin sensitivity [102,103]. It operates through direct activation of G protein-coupled receptors (GPRs) of the class B family and indirect nonreceptor-mediated pathways to protect the liver from NASH progression. Notably, Li et al. [104] observed a significant increase in GLP-1R protein expression in the duodenum and liver of NAFLD rats after eugenol administration, which may be inconsistent with prior research. In obese rats, reduced protein expression of both β-catenin and GLP-1R exacerbates hepatic steatosis induced by fatty acids in rat hepatocytes [37,105]. Conversely, Gao et al. [94] reported that β-catenin and GLP-1R improved steatohepatitis caused by a high-fructose diet. Additionally, a prior study of HFD-induced NASH demonstrated a significant increase in β-catenin expression, which is potentially linked to impaired glucose metabolism [106]. The ameliorative ability of LC and/or GB may be linked to the LC and/or GB antitoxic characteristics provided by their capacities to control oxidative stress based on the antioxidant capacity that originated through their phytoconstituents and to regulate the body response against the impact caused by an HCD. Several findings, such as decreased lipid profiles, glucose levels, insulin resistance, oxidative stress and increased antioxidant activity, support these findings. Additionally, light microscopic analysis of histopathological liver sections revealed a noticeable improvement in the liver structure as well as the immunohistochemical expression of the GLP-1 and β-catenin proteins. TEM analysis, which revealed the restoration of the hepatocyte ultrastructure, supported the beneficial impact of LC and/or GB. All of these effects produced substantial reductions in body weight and serum liver enzymes in the LC and/or GB groups compared with those in the HCD group.

## 5. Conclusions

The outcomes of this work have valuable clinical importance, as LC and GB supplements act as hepatoprotective, hypoglycemic, and antidyslipidemic agents against HCD disorders. Concurrent supplementation with both LC and GB exhibited enhanced efficacy in alleviating liver damage induced by an HCD, surpassing the effects of individual supplements. These valuable effects could occur through ROS scavenging, inhibition of free radical production, antioxidant reactivation, and upregulation of hepatic β-catenin and GLP-1R expression. Therefore, consuming GB with LC supplements can benefit overall health and potentially protect against and treat disorders associated with hyperlipidemia.

## Figures and Tables

**Figure 1 cells-13-00732-f001:**
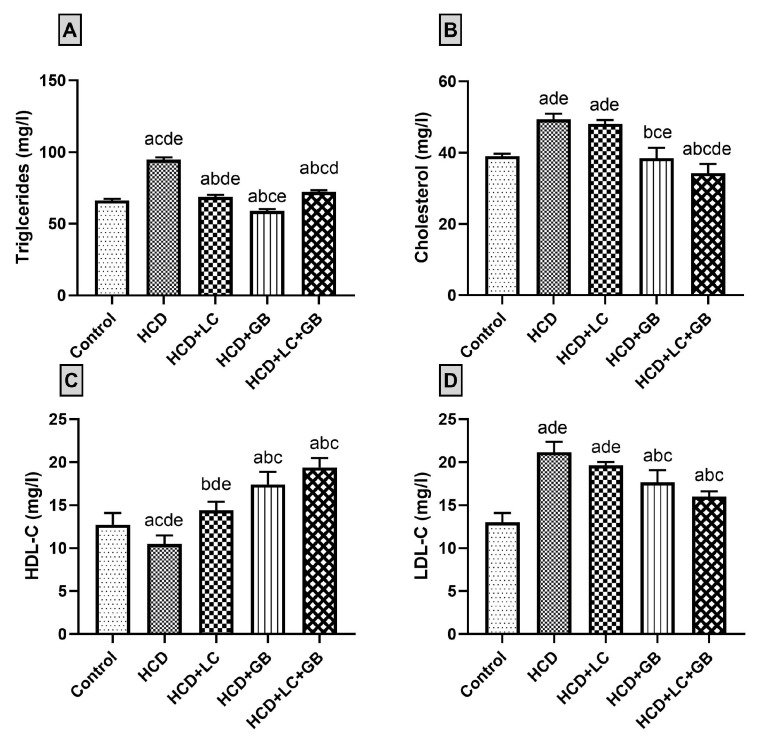
(**A**–**D**) Lipid profile charts. All values are expressed as the mean ± SD (*n* = 6). Triglycerides (**A**), cholesterol (**B**), HDL-C ((**C**) high-density lipoprotein cholesterol), and LDL-C ((**D**) low-density lipoprotein cholesterol) are shown. The letters a–d and e indicate significant differences (*p*-value < 0.05) between groups in comparison with the control, HCD, HCD + LC, HCD + GB, and HCD + LC + GB groups, respectively.

**Figure 2 cells-13-00732-f002:**
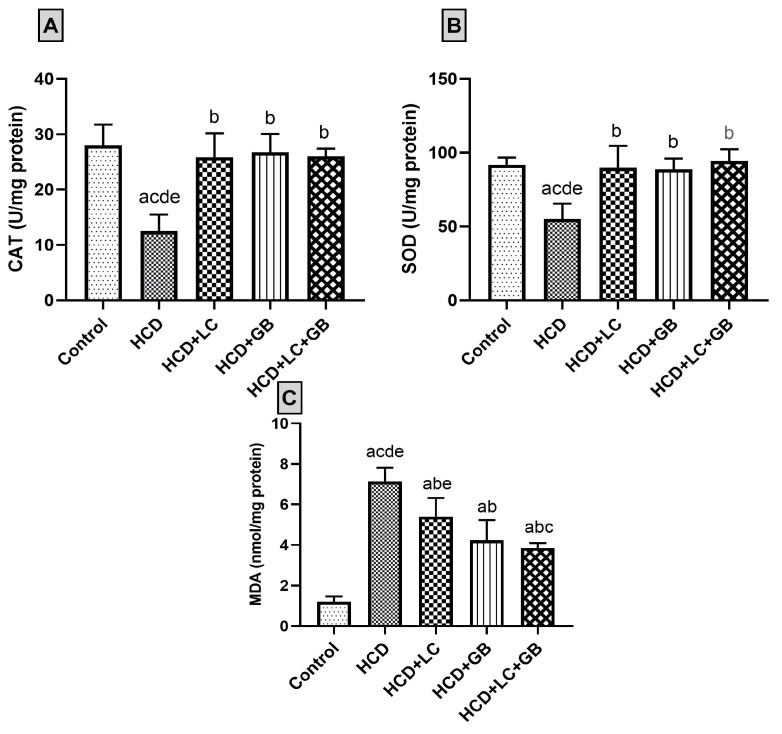
(**A**–**C**) Antioxidant status charts. All values are expressed as the mean ± SD (*n* = 6). Catalase ((**A**); CAT), superoxide dismutase ((**B**); SOD), and malonaldehyde ((**C**); MDA) activities. The letters a–d and e indicate significant differences (*p*-value < 0.05) between groups in comparison with the control, HCD, HCD + LC, HCD + GB, and HCD + LC + GB groups, respectively.

**Figure 3 cells-13-00732-f003:**
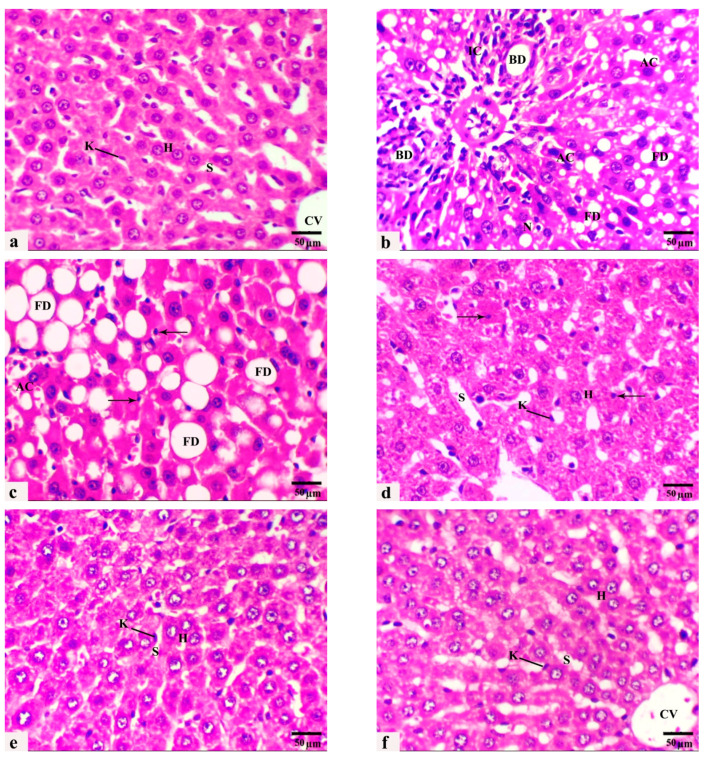
(**a**–**f**): Photomicrographs obtained from H. and E. liver sections of (**a**) control rats displaying normal radial cords of hepatocytes (H), central vein (CV), sinusoid (S), and Kupffer cell (K). (**b**,**c**) HCD-fed rats showing microvesicular steatosis with inflammatory cell infiltration (IC), bile ductulus proliferation (BD), fatty degeneration of swollen hepatocytes with large coalesced vacuoles (FD), apoptotic cell (AC), necrotic area (N), and pyknotic nuclei (thin arrows). (**d**) rats that received HCD + L-carnitine showed a moderate improvement in hepatic architecture. Hepatocytes (H), dilated sinusoids (S), Kupffer cells (K), and pyknotic nuclei (thin arrows). (**e**) rats that received HCD + *Ginkgo biloba* displayed an obvious improvement in hepatic architecture, hepatocytes (H), dilated sinusoids (S), and a few activated Kupffer cells (K). (**f**) Rats that received HCD + LC + GB showed nearly normal hepatic strand architecture, central vein (CV), hepatocytes (H), sinusoids (S), and Kupffer cells (K). H. and E. stain, magnification ×40, scale bar 50 µm.

**Figure 4 cells-13-00732-f004:**
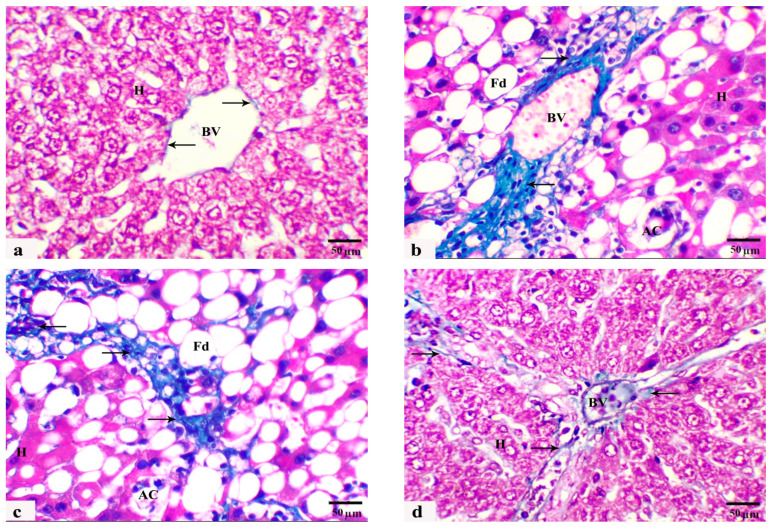
(**a**–**f**) Photomicrographs obtained from Masson trichrome liver sections of rats from the control ((**a**)/G1), HCD ((**b**,**c**)/G2), HCD + LC ((**d**)/G3), HCD + GB ((**e**)/G4), and HCD + LC + GB ((**f**)/G5) groups showing positive blue collagen fiber deposition (arrows), hepatocytes (H), apoptotic cells (AC), and blood vessels (BV). Masson’s trichrome stain, 40×; scale bar, 50 µm. (**g**) The graph shows the semiquantitative scoring of the percentage of positive staining. The results are presented as the mean ± SD, with a sample size of *n =* 6. The letters on columns a–e indicate significant differences (*p*-value < 0.05) between groups and the control, HCD, HCD + LC, HCD + GB, and HCD + LC + GB groups, respectively. MT stain, magnification 40×, scale bar 50 µm.

**Figure 5 cells-13-00732-f005:**
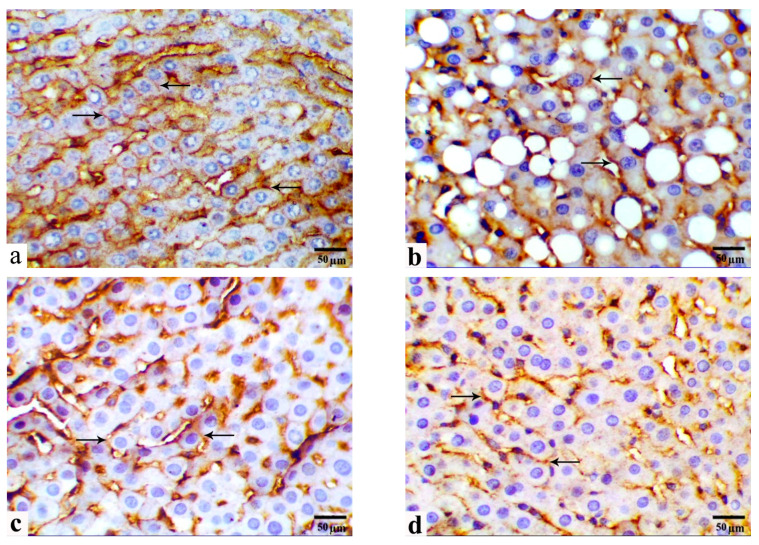
(**a**–**f**) Photomicrographs obtained from IHC liver sections of control rats ((**a**)/G1), HCD-fed rats ((**b**)/G2), HCD + LC-fed rats ((**c**)/G3), HCD + GB-fed rats ((**d**)/G4), and HCD + LC + GB-fed rats ((**e**)/G5) showing β-catenin expression (arrows). Scale bar, 50 µm. (**f**) The graph shows the semiquantitative scoring of the percentage of positive staining; the results are presented as the mean ± SD, with a sample size of *n =* 6. The letters on columns a–e indicate significant differences (*p*-value < 0.05) between groups and the control, HCD, HCD + LC, HCD + GB, and HCD + LC + GB groups, respectively. IHC, magnification 40×, scale bar 50 µm.

**Figure 6 cells-13-00732-f006:**
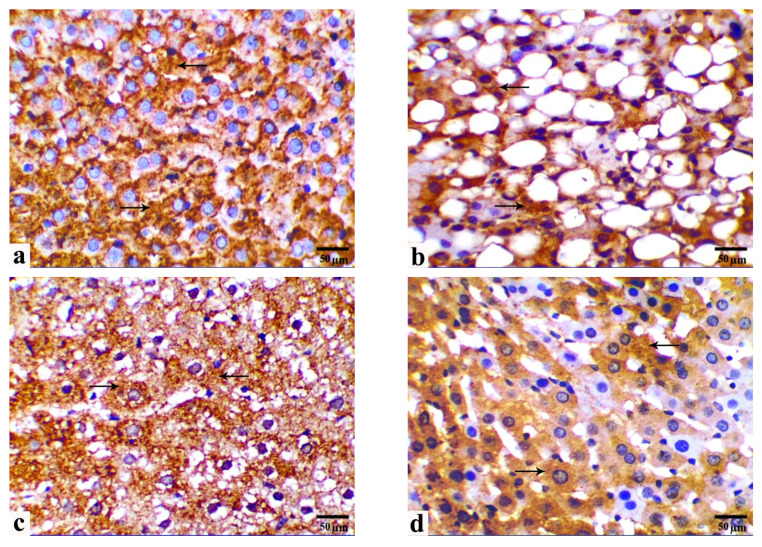
(**a**–**f**) Photomicrographs obtained from IHC liver sections of control rats ((**a**)/G1), HCD-fed rats ((**b**)/G2), HCD + LC-fed rats ((**c**)/G3), HCD + GB-fed rats ((**d**)/G4), and HCD + LC + GB-fed rats ((**e**)/G5) showing glucagon-like peptide-1 expression (arrows). Scale bar, 50 µm. (**f**) The graph shows the semiquantitative scoring of the percentage of positive staining; the results are presented as the mean ± SD, with a sample size of *n =* 6. The letters on columns a–e indicate significant differences (*p*-value < 0.05) between groups and the control, HCD, HCD + LC, HCD + GB, and HCD + LC + GB groups, respectively. IHC, magnification 40×, scale bar 50 µm.

**Figure 7 cells-13-00732-f007:**
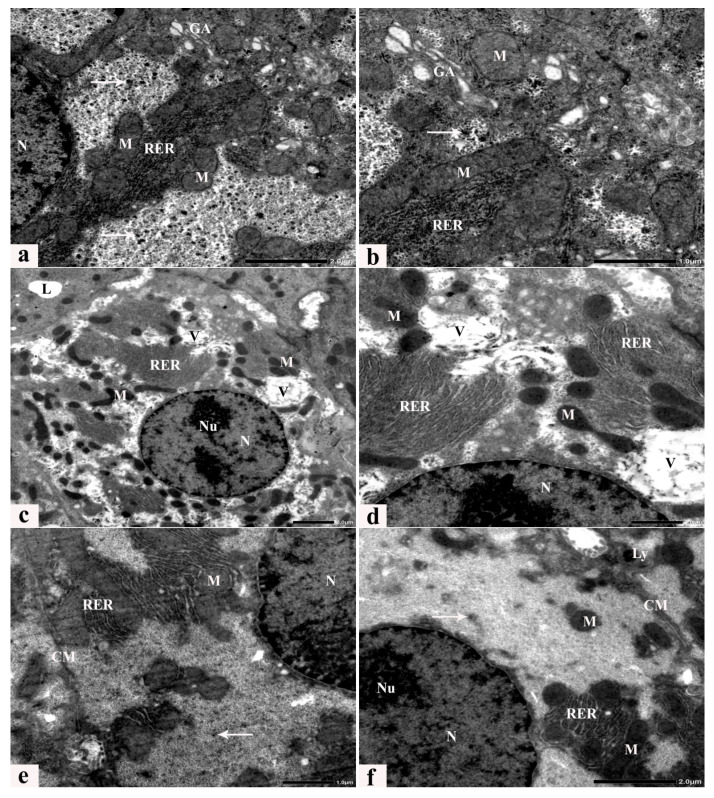
(**a**–**h**) Electron micrographs of a hepatocyte from a control rat (**a**,**b**), an HCD-fed rat (**c**,**d**), and animals that received HCD + LC (**e**), HCD + GB (**f**), and HCD + LC + GB (**g**,**h**) showing the cell membrane (CM), nucleus (N), nucleolus (NU), rough endoplasmic reticulum (rER), Golgi apparatus (GA), mitochondria (M), cristae (C), glycogen rosettes (arrows), lysosomes (LY), degenerated lytic cytoplasm with large vacuoles (V), Uranvl acetate, and lead citrate stain.

**Table 1 cells-13-00732-t001:** Effects of LC and/or GB treatment on body weight, serum glucose homeostasis, and liver enzymes.

Groups	Body Weight	Glucose (mg/L)	Insulin (uIU/mL)	HOMA-IR	ALT(U/L)	AST (U/L)
**Control**	157.67 ± 5.28	86.17 ± 8.1	16.67 ± 1.15	3.73 ± 0.39	48.17 ± 4.67	94.5 ± 9.42
**HCD**	259.83 ± 5.27 ^acde^	138.22 ± 4.22 ^acde^	21.6 ± 3.51 ^e^	5.1 ± 0.55 ^a^	91.67 ± 4.18 ^acde^	166 ± 11.08 ^acde^
**HCD + LC**	218.83 ± 3.43 ^abe^	119.12 ± 4.02 ^abe^	18.83 ± 2.52	4.88 ± 0.46 ^a^	62.5 ± 5.86 ^ab^	118 ± 8.74 ^ab^
**HCD + GB**	228 ± 7.48 ^abe^	119.28 ± 4.83 ^abe^	16.63 ± 1.31	4.78 ± 0.48 ^a^	62 ± 5.18 ^ab^	118.33 ± 5.47 ^abc^
**HCD + LC + GB**	191.83 ± 5.19 ^abcd^	94.76 ± 3.79 ^bcd^	15.23 ± 1.68 ^b^	4.087 ± 0.44	55 ± 4 ^b^	113.33 ± 3.72 ^ab^

All values are expressed as the mean ± SD; HOMA-IR: homeostatic model assessment for insulin resistance; ALT: alanine aminotransferase; AST: aspartate aminotransferase. The superscript letters a, b, c, d, and e indicate significant differences (*p*-value < 0.05) between groups in comparison with the control, HCD, HCD + LC, HCD + GB, and HCD + LC + GB groups, respectively.

**Table 2 cells-13-00732-t002:** Quantitative grading and scoring system for nonalcoholic steatohepatitis-related histopathological lesions in the livers of rats in different groups.

Groups	Control (Standard Diet)	HCD	HCD + LC	HCD + GB	HCD + LC + GB
Lesions					
**Hepatic steatosis**	0.00 ± 0.00 ^b^	2.70 ± 0.00 ^a^	1.00 ± 0.00 ^ab^	1.20 ± 0.00 ^ab^	0.50 ± 0.00 ^b^
**Ballooning degeneration**	0.00 ± 0.00 ^b^	2.00 ± 0.00 ^a^	1.00 ± 0.00 ^ab^	0.60 ± 0.00 ^ab^	0.20 ± 0.00 ^b^
**Lobular inflammation**	0.00 ± 0.00 ^b^	2.00 ± 0.00 ^a^	0.60 ± 0.00 ^ab^	0.40 ± 0.00 ^b^	0.10 ± 0.00 ^b^
**Hepatic fibrosis**	0.00 ± 0.00 ^b^	3.00 ± 0.00 ^a^	1.00 ± 0.00 ^a^	0.50 ± 0.00 ^b^	0.30 ± 0.00 ^b^
**Steatosis grade**; 5% (0), 5–33% (1), 34–66% (2), 66% (3).					
**Hepatocellular ballooning**: none (0), little ballooned hepatocytes (1), several ballooned hepatocytes (2).					
**Lobular inflammation**: no foci (0), 2 foci (1), 2–4 foci (2), 4 foci (3) per 20× field.					
**Hepatic fibrosis**: No fibrosis (0), fibrous with or without short septa in some portal areas (1), fibrous with or without short septa in most of the portal tracts (2), fibrous with portal-portal bridges in most portal areas (3), fibrous with portal-portal bridges and portal enter bridges in portal areas (4), noticeable portal-portal bridges or/and bridges with occasional nodules (5), and cirrhosis (6).					

Values are the mean ± SD. Values (*p* < 0.05) followed by different superscript letters, a and b, for the vertical row are significantly different between all groups in comparison with the control and HCD groups.

## Data Availability

All data used in this study are included in this published article.

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
