# Peer review of "L-carnitine and Ginkgo biloba Supplementation In Vivo Ameliorates HCD-Induced Steatohepatitis and Dyslipidemia by Regulating Hepatic Metabolism"

_cells, 2024, doi:10.3390/cells13090732_

Round 1

Reviewer 1 Report

Comments and Suggestions for Authors

The study explored the potential therapeutic mechanism of the dietary intake L-carnitine (LC) and Ginkgo biloba leaf extract (GB) supplementation in ameliorating the adverse effects of hyperlipidaemia and hepatosteatosis induced by a high-cholesterol diet (HCD) in rat model. The study did offer some evidence that dietary LC+GB presents the alternation of hepatic lipid metabolism in HCD-treated rats. However, I do have some questions raised.

1.   Please provide more basic information about the LC and GB, which may alter the hepatic lipid metabolism in the Introduction section, especially the lipid dysregulation in HCD-induced rats. If the authors did not focus on the regular dietary factors (i.e., LC & GB), a black box targets dietary components (macronutrients) and may not be elucidated.

2. What are the potential components of the extract of GB? Basic information should be provided, such as molecule weight and other physical properties. Otherwise, that would be a different possible mechanism for this HCD rat model.

3. What is the aim of this work that I would like to express? Authors should point out the relationship between the LC and GB in metabolic biochemical parameters and why the LC & GB combination benefits certain hyperlipidemia symptoms, especially in cholesterol metabolism. The authors may provide the original biochemical data to support the idea that LC + GB is a healthy combination that regulates the status of hepatic lipids in this study.

4. In Figure 1, the level of HDLc should be much higher than LDLc in the Wistar albino rat model, especially in the HCD model. Please check it again.

5. Please provide more information on treated LC and/or GB doses in the Method section. Why choose such doses to treat the serum and hepatic hyperlipidemia status? 

6. If HCD-treated rats were ameliorated by the LC and/or GB, such alterations of dysregulation and other biochemical parameters (NASH-related inflammatory biomarkers) should be provided. The possible mechanism should also be discussed in the text. However, I did not find such information.

7. As a result, some of the data in the Table should be more accurate in scientific format. Group names should be easy to follow, not just G1 or G5. More clear photography should be provided in Figure 7.

8. Overall, this was an important topic, but there was no direct evidence to demonstrate that LC and/or GB ameliorated hepatic dysfunction. More solid data and text modifications are necessary. Thus, I do not recommend it in current status.

Reviewer 2 Report

Comments and Suggestions for Authors

In this study, the authors showed that a high-cholesterol diet (HCD)-induced hyperlipidemia and liver steatosis were attenuated by the administration of L-carnitine (LC), Ginkgo biloba (GB), or their combination. The results are meaningful and suggest that LC and GB could be supplements to improve lipid metabolism and overall health. I would like to raise some points that should be improved by the authors. 

- Title: Provide the extension for HCD.

- Abstract: Provide the doses of L-carnitine (LC) and Ginkgo biloba (GB) that were used.

- 2.2. Animals and treatment: Include the age and the keeping conditions of the animals. Also, why the use of only male animals?

- 2.3. Serum biochemical analysis: Here, the authors also include the oxidative stress measurements that were performed, such as MDA, as well as SOD and CAT enzyme activities. For better visualization, I believe this could fit in another subheading and the authors should describe these methods in more detail. Moreover, what about other oxidative stress markers (carbonyl protein, reactive species, …)?

- 2.7. Statistical analysis: Include the statistical program used for comparisons. Also, in the figure legends, provide the values from the statistical analysis (e.g., F(x,x) = xx; n = xx; P = x.xx).

- Section 3. Results (L. 158-160): Remove this part: “This section may be divided by subheadings. It should provide a concise and precise description of the experimental results, their interpretation, as well as the experimental conclusions that can be drawn”.

- Figures: Letters in the graphs used to show differences among groups are difficult to follow, please improve the description or change the comparison parameters. To aid in this layout formatting, represent the groups in different symbols/colors.

- Tables and figure legends: They should be concise although informative. Please include the appropriate statistical results, scale bar lengths, define abbreviations, and add references to any symbols used in the figures.

Reviewer 3 Report

Comments and Suggestions for Authors

Nofal et al. investigated the therapeutic effects of L-carnitine and Ginkgo biloba on hyperlipidemia and hepatosteatosis. The manuscript is well-written, the authors cited recent references. However, in the Materials and methods section, there are several unanswered questions that are essential to understanding the research findings.

Major comments: 

1. How were the LC and GB doses selected? Please, indicate it.

2. How was the 8-week treatment period determined?

3. How were the rats terminated? Please describe the mode of the euthanasia.

4. Have the authors monitored the food consumption?

5. The authors mentioned that one-way ANOVA was selected to statistical analyses. It would improve the MS, if possible interactions between different treatments were also presented.

Minor point:

Please, delete the following part from the Results section:

“This section may be divided by subheadings. It should provide a concise and precise description of the experimental results, their interpretation, as well as the experimental conclusions that can be drawn.”

Reviewer 4 Report

Comments and Suggestions for Authors

This study explored the potential therapeutic mechanisms of L-carnitine and Ginkgo biliba leaf extract applied single and in association in ameliorating the effects of hyperlipidemia and hepatic steatosis induced by a high cholesterol diet in an animal model. The authors revealed that treatment with LC, GB or LC +GB in association reduced serum glucose, insulin, HOMA-IR, total cholesterol, LDL-c, triglycerides, hepatic enzymes levels and increased HDL-c, improved antioxidant ability and reduced lipid peroxidation, and upregulated the immune expression of GLP-1R and B-cathenin expresion.

Did the authors perform multivariable linear regression to study the relationship between treatment with LC plus GB and the other biochemical, immunohistochemical, ultrastructural parameters and markers of oxidative stress? If so, what was the result? 

Round 2

Reviewer 1 Report

Comments and Suggestions for Authors

Authors responded the considerations properly. I have no further questions.

Reviewer 3 Report

Comments and Suggestions for Authors

All questions have been properly answered.